

# A Bayesian Approach to Multivariate Adaptive Localization in Ensemble-Based Data Assimilation with Time-Dependent Extensions

Andrey A Popov[1] and Adrian Sandu[1]

[1]Department of Computer Science, Virginia Tech, 2202 Kraft Drive, Blacksburg, VA 24060

**Correspondence:** Andrey A Popov (apopov@vt.edu)

**Abstract.** Ever since its inception, the Ensemble Kalman Filter has elicited many heuristic methods that sought to correct it. One such method is localization—the thought that 'nearby' variables should be highly correlated with 'far away' variable not. Recognizing that correlation is a time-dependent property, adaptive localization is a natural extension to these heuristics. We propose a Bayesian approach to adaptive Schur-product localization for the DEnKF, and extend it to support multiple radii of influence. We test both the empirical validity of (multivariate) adaptive localization, and of our approach. We test a simple toy problem (Lorenz'96), extending it to a multivariate model, and a more realistic geophysical problem (1.5 Layer Quasi-Geostrophic). We show that the multivariate approach has great promise on the toy problem, and that the univariate approach leads to improved filter performance for the realistic geophysical problem.

## 1 Introduction

We consider a computational model that approximates the evolution of a physical dynamical system such as the atmosphere:

$$\mathbf{x}_{i+1} = \mathcal{M}_{i,i+1}(\mathbf{x}_i) + \boldsymbol{\xi}_{i,i+1}, \tag{1}$$

The (finite-dimensional) state of the model $\mathbf{x}_i \in \mathbb{R}^n$ at time $t_i$ approximates the (finite-dimensional projection of the) physical true state $\mathbf{x}_i^{\mathrm{t}} \in \mathbb{R}^n$. The computational model (1) is inexact, and we assume the model error to be additive and normally distributed, $\boldsymbol{\xi}_{i,i+1} \sim \mathcal{N}(\mathbf{0}, \mathbf{Q}_i)$.

The initial state of the model is also not precisely known, and to model this uncertainty we consider that it is a random variable drawn from a specific probability distribution:

$$\mathbf{x}_0 \sim \mathcal{N}(\mathbf{x}_0^{\mathrm{t}}, \mathbf{P}_0^{\mathrm{b}}). \tag{2}$$

Consider also observations of the truth,

$$\mathbf{y}_{i+1} = \mathcal{H}(\mathbf{x}_{i+1}^{\mathrm{t}}) + \boldsymbol{\eta}_{i+1}, \tag{3}$$

that are corrupted by normal observation errors $\boldsymbol{\eta}_{i+1} \sim \mathcal{N}(\mathbf{0}, \mathbf{R}_{i+1})$. We consider here the case with a linear observation operator, $\mathcal{H} := \mathbf{H}$.





Data assimilation (Asch et al., 2016; Law et al., 2015; Evensen, 2009; Reich and Cotter, 2015) fuses information from the model forecast states (1) and observations (3) in order to obtain an improved estimation of the truth at any given point in time. Data assimilation approaches include the Ensemble Kalman Filters (EnKF) (Evensen, 1994, 2009; Constantinescu et al., 2007b) that rely on Gaussian assumptions, particle filters for non-Gaussian distributions (Van Leeuwen et al., 2015; Attia et al.,
2017; Attia and Sandu, 2015), and variational approaches, rooted in control theory (Dimet and Talagrand, 1986; Sandu et al., 2005; Carmichael et al., 2008)

The EnKF is an important family of data assimilation techniques that propagate both the mean and covariance of the state uncertainty (2) through the model (1) using a Monte-Carlo approach. While large dynamical systems of interest have a large number of modes along which errors can grow, the number of ensemble members used to characterize uncertainty remains
relatively small due to computational costs. As a result, inaccurate correlation estimates obtained through Monte Carlo sampling can profoundly affect the filter results. Techniques such as covariance localization and inflation have been developed to alleviate these problems (Petrie and Dance, 2010).

Localization techniques take advantage of the fundamental property of geophysical systems that correlations between variables decrease with spatial distance (Kalnay, 2003; Asch et al., 2016). This prior knowledge is used to scale ensemble-estimated
covariances between distant variable such as to reduce inaccurate, spurious correlations. The prototypical approach to localization is a Schur-product-based tapering of the covariance matrix (Bickel et al., 2008); theoretical results ensure that the covariance matrices estimated using small ensembles sampled from a multivariate normal probability distribution, upon tapering, approach quickly the true covariance matrix. Practical implementations of localization rely on restricting the information flow, either in state space or in observation space, to take place within a given "radius of influence" (Hunt et al., 2007). Some
variants of EnKF like the Maximum Likelihood Ensemble Filter (MLEF) (Zupanski, 2005) reduce the need for localization, while others use localization in order to efficiently parallelize in space the analysis cycles (Nino-Ruiz et al., 2015).

The performance of the EnKF algorithms critically depends on the correct choice of localization radii (a.k.a, the decorrelation distances), since values that are too large fail to correct for spurious correlations, while values that are too small throw away important correlation information. However, the physical values of the spatial decorrelation scales are not known apriori, and
they change with the temporal and spatial location. At the very least the decorrelation scales depend on the current atmospheric flow. In atmospheric chemistry systems, because of the drastic difference in reactivity, each chemical species has its own individual localization radius (Constantinescu et al., 2007a). Clearly, the widely used approach of estimating decorrelation distances from historical ensembles of simulations, and then using a constant averaged value throughout the space and time domain, lead to a suboptimal performance of Ensemble Kalman filters.

Adaptive localization schemes seek to estimate decorrelation distances from the data, such as to optimize the filter performance according to some criteria. One approach to adaptive localization utilizes an ensemble of ensembles to detect and mitigate spurious correlations (Anderson, 2007). Relying purely on model dynamics and foregoing reliance on spatial properties of the model, the method is very effective for small scale systems, but its applicability to large-scale geophysical problems is unclear. There is, however, evidence that optimal localization depends more on ensemble size and observation properties
than on model dynamics (Kirchgessner et al., 2014), and that adaptive approaches whose correlation functions follow these



dynamics do not show a significant improvement over conventional static localization (Bishop and Hodyss, 2011). Methods such as sampling error correction (Anderson, 2012) take advantage of these properties to build correction factors and apply them as an ordinary localization scheme. A different approach uses the inherent properties of correlation matrices to construct Smoothed ENsemble COrrelations Raised to a Power (SENCORP) (Bishop and Hodyss, 2007) matrices, that in the limiting

case remove all spurious correlations. This method relies purely on the statistical properties of correlation matrices, and ignores the model dynamics and the spatial and temporal dependencies it defines. A recent approach considers machine learning algorithms to capture hidden properties in the propagation model that affect the localization parameters (Moosavi et al., 2018).

This work develops a Bayesian framework to dynamically learn the parameters of the Schur-product based localization from the ensemble of model states and the observations during the data assimilation in geophysical systems. Specifically, the

localization radii are considered random variables described by parametrized distributions, and are retrieved as part of the assimilation step together with the analysis states.

The paper is organized as follows. Sect. 2 reviews background material for Ensemble Kalman filtering and Schur-product localization. Sect. 3 describes the proposed theoretical framework for localization and the resulting optimization problems for maximum likelihood solutions. Numerical results with three test problems reported in Sect. 4 provide empirical validation of

the proposed approach.

## 2  Background

Consider and ensemble of $N$ states $\mathbf{x} \in \mathbb{R}^{n \times N}$ sampling the probability density that describes the uncertainty in the state at a given time moment (the time subscripts are omitted for simplicity of notation). The ensemble mean, ensemble anomalies, and ensemble covariance are, respectively:

$$20 \quad \bar{\mathbf{x}} = \frac{1}{N}\mathbf{x}\mathbf{1}_N, \quad \mathbf{X} = \mathbf{x} - \bar{\mathbf{x}}\mathbf{1}_N^{\mathsf{T}}, \quad \mathbf{P} = \frac{1}{N-1}\mathbf{X}\mathbf{X}^{\mathsf{T}}. \tag{4}$$

Typically $N$ is smaller than the number of positive Lyapunov exponents in our dynamical system, and much smaller than the number of states (Bergemann and Reich, 2010). Consequently, the ensemble statistics (4) are marred by considerable sampling errors. The elimination of sampling errors, manifested as spurious correlations in the covariance matrix (Evensen, 2009), leads to the need for localization.

### 25  2.1  Kalman analysis

The mean and the covariance are propagated first through the forecast step. Specifically, each ensemble member is advanced to the current time using the model (1) to obtain the ensemble forecast $\mathbf{x}^{\mathrm{f}}$ (with mean $\bar{\mathbf{x}}^{\mathrm{f}}$ and covariance $\mathbf{P}^{\mathrm{f}}$) at the current time. A covariance inflation step $\mathbf{X}^{\mathrm{f}} \leftarrow \alpha\mathbf{X}^{\mathrm{f}}, \alpha > 1$, can be applied to prevent filter divergence (Anderson, 2001).

The mean and covariance are then propagated through the analysis step, which fuses information from the forecast mean

30  and covariance and from observations (3), to provide an analysis ensemble $\mathbf{x}^{\mathrm{a}}$ (with mean $\bar{\mathbf{x}}^{\mathrm{a}}$ and covariance $\mathbf{P}^{\mathrm{a}}$) at the same



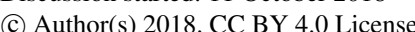



time. Here we consider the deterministic EnKF (DEnKF) (Sakov and Oke, 2008) which obtains the analysis as follows:

$$\bar{\mathbf{x}}^{a} = \bar{\mathbf{x}}^{f} + \mathbf{K}\mathbf{d}, \tag{5a}$$

$$\mathbf{K} = \mathbf{P}^{f}\mathbf{H}^{\intercal}\left(\mathbf{H}\mathbf{P}^{f}\mathbf{H}^{\intercal} + \mathbf{R}\right)^{-1}, \tag{5b}$$

$$\mathbf{d} = \mathbf{y} - \mathbf{H}\bar{\mathbf{x}}^{f}, \tag{5c}$$

$$\mathbf{X}^{a} = \mathbf{X}^{f} - \frac{1}{2}\mathbf{K}\mathbf{H}\mathbf{X}^{f}, \tag{5d}$$

$$\mathbf{x}^{a} = \bar{\mathbf{x}}^{a}\mathbf{1}_{N}^{\intercal} + \mathbf{X}^{a}, \tag{5e}$$

where $\mathbf{K}$ is the Kalman gain matrix and $\mathbf{d}$ the vector of innovations. DEnKF is chosen for simplicity of implementation and ease of amending to support Schur product-based localization. However, the adaptive localization techniques discussed herein are general—they do not depend on this choice and are equally applicable to any EnKF algorithm.

## 2.2 Schur product localization

Covariance localization involves the Schur (element-wise) product between a symmetric positive semi-definite matrix $\boldsymbol{\rho}$ and the ensemble forecast covariance:

$$\mathbf{P}^{f} \leftarrow \boldsymbol{\rho} \circ \mathbf{P}^{f}, \quad \mathbf{P}_{i,j}^{f} \leftarrow \rho_{i,j}\,\mathbf{P}_{i,j}^{f}. \tag{6}$$

By the Schur product theorem (Schur, 1911), if $\boldsymbol{\rho}$ and $\mathbf{P}^{f}$ are positive semi-definite, then their Schur product is positive semi-

15 definite. The matrix $\boldsymbol{\rho}$ is chosen such that it reduces the sampling errors and brings the ensemble covariance closer to the true covariance matrix. We note that one can apply the localization in ways that mitigate the problem of storing full covariance matrices (Houtekamer and Mitchell, 2001). Efficient implementation aspects are not discussed further as they do not impact the adaptive localization approaches developed herein.

We seek to generate the entries of localization matrix $\boldsymbol{\rho}$ from a localization function $\ell : \mathbb{R}_{\geq 0} \rightarrow [0, 1]$, used to represent the

20 weights applied to each individual covariance:

$$\boldsymbol{\rho} = [\ell(d(i,j)/r)]_{1 \leq i,j \leq n}. \tag{7}$$

The function $\ell$ could be thought of as a regulator which brings the ensemble correlations in line with the physical correlations, which are often compactly supported functions (Gneiting, 2002). The metric $d$ quantifies the physical distance between model variables, such that $d(i,j)$ represents the spatial distance between the state-space variables $x_i$ and $x_j$. The "radius of influence"

parameter $r$ represents the correlation spatial scale; the smaller $r$ is the faster variables decorrelate with increasing distance.

If the spatial discretization is time-invariant, and $\ell$ and $r$ are fixed, that the matrix $\boldsymbol{\rho}$ is also time invariant. The goal of the adaptive localization approach is to learn the best value of $r$ dynamically from the ensemble.

A common localization function used in production software is due to Gaspari and Cohn (Gaspari and Cohn, 1999; Gneiting, 2002; Petrie, 2008). Here we use the Gaussian function

$$\ell(u) = \exp\left(-u^2/2\right) \tag{8}$$

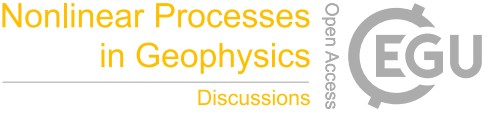
to illustrate the adaptive localization strategy. However, our approach is general and can be used with any suitable localization function.

## 3 Bayesian Approach

We denote the analysis step of the filter by:

$$\mathbf{x}^{\mathrm{a}} = \mathcal{A}(\mathbf{x}^{\mathrm{f}}, \mathbf{y}, \boldsymbol{\upsilon}), \tag{9}$$

where $\boldsymbol{\upsilon}$ are the localization and inflation parameters. In this paper we look solely at localization. We consider here state-space localization and define the mapping operator $\mathfrak{r} : \mathbb{N}_n \to \mathbb{N}_g$ that assigns each state-space component to a group. All state variables assigned to the same group are analyzed using the same localization radius. Assuming a fixed mapping, the localization parameters are the radii values for each group $\boldsymbol{\upsilon} \in \mathbb{R}^g$. These values can be tuned independently during the adaptive algorithm. The corresponding prolongation operator $\mathfrak{p} : \mathbb{R}^g \to \mathbb{R}^n$ assigns one of the $g$ radii values to each of the $n$ state-space components:

$$\mathbf{r}_{i+1} = \mathfrak{p}(\boldsymbol{\upsilon}_{i+1}). \tag{10}$$

For univariate localization $g = 1$ and $\mathfrak{p}(\upsilon) = \upsilon \mathbf{1}_n$.

In the Bayesian framework, we consider the localization parameters to be random variables with an associated prior probability distribution. Specifically, we assume that each radii $\upsilon^{(j)}$ is independently distributed according to a univariate gamma distribution $\upsilon^{(j)} \sim \Gamma(\alpha^{(j)}, \beta^{(j)})$. The gamma probability density:

$$f_\Gamma(\upsilon; \alpha, \beta) = \frac{\beta^\alpha \upsilon^{\alpha-1} e^{-\beta \upsilon}}{\Gamma(\alpha)}, \quad \upsilon, \alpha, \beta > 0 \tag{11}$$

has mean $\bar{\upsilon} = \alpha/\beta$ and variance $\mathrm{Var}(\upsilon) = \alpha/\beta^2$. We have chosen the gamma distribution as it is the maximum entropy probability distribution for a fixed mean (Singh et al., 1986), i.e., is the best distribution that one can choose without additional information. It is supported on the interval $(0, \infty)$ and has exactly two free parameters (allowing to control both the mean and variance).

The assimilation algorithm computes a posterior (analysis) probability density over the state space considering the probabilities of observations and parameters. We start with Bayes' identity:

$$\pi(\mathbf{x}, \boldsymbol{\upsilon} \mid \mathbf{y}) \propto \pi(\mathbf{y} \mid \mathbf{x}, \boldsymbol{\upsilon}) \, \pi(\mathbf{x} \mid \boldsymbol{\upsilon}) \, \pi(\boldsymbol{\upsilon})$$
$$= \pi(\mathbf{y} \mid \mathbf{x}, \boldsymbol{\upsilon}) \, \pi(\mathbf{x} \mid \boldsymbol{\upsilon}, \mathbf{y}) \, \pi(\boldsymbol{\upsilon}) \, \pi(\mathbf{y}). \tag{12}$$

Note that $\pi(\mathbf{y})$ is a constant scaling factor. Here $\pi(\boldsymbol{\upsilon})$ represents the (prior) uncertainty in the parameters, $\pi(\mathbf{x} \mid \mathbf{y}, \boldsymbol{\upsilon})$ represents the uncertainty in the state for a specific value of the parameters, and $\pi(\mathbf{y} \mid \mathbf{x}, \boldsymbol{\upsilon})$ represents the likelihood of observations with respect to both state and parameters. Under Gaussian assumptions on the state and observation errors equation (12) can be





explicitly written out as:

$$\pi(\mathbf{x}, \boldsymbol{v} \mid \mathbf{y}) \propto \exp\left(-\frac{1}{2}\left\|\mathcal{A}(\mathbf{x}^{\mathrm{f}}, \mathbf{y}, \boldsymbol{v}) - \mathbf{x}^{\mathrm{f}}\right\|^2_{\mathbf{P}^{\mathrm{f}}_{(\boldsymbol{v})}{}^{-1}}\right)$$
$$\exp\left(-\frac{1}{2}\left\|\mathbf{y} - \mathbf{H}\mathcal{A}(\mathbf{x}^{\mathrm{f}}, \mathbf{y}, \boldsymbol{v})\right\|^2_{\mathbf{R}^{-1}}\right)$$
$$f_{\mathcal{P}}(\boldsymbol{v}), \tag{13}$$

with $f_{\mathcal{P}}(\boldsymbol{v})$ given by equation (11). Note that, once the analysis scheme (9) has been chosen, the analysis state becomes a

5   function of $\boldsymbol{v}$, and the conditional probability in (13) represents the posterior probability of $\boldsymbol{v}$.

The negative log likelihood of the posterior probability (13) is:

$$\mathcal{J}(\boldsymbol{v}) = -\log\left[\exp\left(-\frac{1}{2}\left\|\mathcal{A}(\mathbf{x}^{\mathrm{f}}, \mathbf{y}, \boldsymbol{v}) - \mathbf{x}^{\mathrm{f}}\right\|^2_{\mathbf{P}^{\mathrm{f}}_{(\boldsymbol{v})}{}^{-1}}\right)\right.$$
$$\exp\left(-\frac{1}{2}\left\|\mathbf{y} - \mathbf{H}\mathcal{A}(\mathbf{x}^{\mathrm{f}}, \mathbf{y}, \boldsymbol{v})\right\|^2_{\mathbf{R}^{-1}}\right)$$
$$\left.\prod_{j=1}^{g} v^{(j), \alpha - 1}\exp\left(-\beta^{(j)}v^{(j)}\right)\right]$$
$$= \tfrac{1}{2}\left\|\mathcal{A}(\mathbf{x}^{\mathrm{f}}, \mathbf{y}, \boldsymbol{v}) - \mathbf{x}^{\mathrm{f}}\right\|^2_{\mathbf{P}^{\mathrm{f}}_{(\boldsymbol{v})}{}^{-1}}$$
$$+ \tfrac{1}{2}\left\|\mathbf{y} - \mathbf{H}\mathcal{A}(\mathbf{x}^{\mathrm{f}}, \mathbf{y}, \boldsymbol{v})\right\|^2_{\mathbf{R}^{-1}}$$
$$+ \sum_{j=1}^{g}\left(\beta^{(j)}v^{(j)} - \left(\alpha^{(j)} - 1\right)\log\left(v^{(j)}\right)\right). \tag{14}$$

The gradient of the cost function with respect to individual localization parameters reads:

$$\frac{\partial \mathcal{J}}{\partial v^{(j)}} = \tfrac{1}{2}\left\|\mathbf{P}^{\mathrm{f}}_{(\boldsymbol{v})}{}^{-1}\left(\mathcal{A}(\mathbf{x}^{\mathrm{f}}, \mathbf{y}, \boldsymbol{v}) - \mathbf{x}^{\mathrm{f}}\right)\right\|^2_{\frac{\partial \mathbf{P}^{\mathrm{f}}_{(\boldsymbol{v})}}{\partial v^{(j)}}}$$
$$+ \frac{\partial \mathcal{A}(\mathbf{x}^{\mathrm{f}}, \mathbf{y}, \boldsymbol{v})}{\partial v^{(j)}}^{\mathsf{T}}\mathbf{P}^{\mathrm{f}}_{(\boldsymbol{v})}{}^{-1}\left(\mathcal{A}(\mathbf{x}^{\mathrm{f}}, \mathbf{y}, \boldsymbol{v}) - \mathbf{x}^{\mathrm{f}}\right)$$
$$- \left(\mathbf{H}\frac{\partial \mathcal{A}(\mathbf{x}^{\mathrm{f}}, \mathbf{y}, \boldsymbol{v})}{\partial v^{(j)}}\right)^{\mathsf{T}}\mathbf{R}^{-1}\left(\mathbf{y} - \mathbf{H}\mathcal{A}(\mathbf{x}^{\mathrm{f}}, \mathbf{y}, \boldsymbol{v})\right)$$
$$+ \beta^{(j)} - (\alpha^{(j)} - 1)\frac{1}{v^{(j)}}, \tag{15}$$

where we took advantage of the properties of symmetric matrices. Note that without the assumption that parameters are independent the gradient would involve higher order tensors.



### 3.1 Solving the optimization problem

Under the assumptions that the analysis function (9) is based on DEnKF (5), and that all ensemble members are i.i.d., we obtain:

$$
\left\| \mathcal{A}(\mathbf{x}^\mathrm{f}, \mathbf{y}, \boldsymbol{v}) - \mathbf{x}_i^\mathrm{f} \right\|_{\mathbf{P}_{(\boldsymbol{v})}^{\mathrm{f}-1}}^2 = \sum_{e=1}^{N} \left\| \mathbf{K}_{(\boldsymbol{v})} \mathbf{z}^{(e)} \right\|_{\mathbf{P}_{(\boldsymbol{v})}^{\mathrm{f}-1}}^2
$$

$$
= \sum_{e=1}^{N} \left\| \mathbf{S}_{(\boldsymbol{v})}^{-1} \mathbf{z}^{(e)} \right\|_{\mathbf{H}\mathbf{P}_{(\boldsymbol{v})}^{\mathrm{f}} \mathbf{H}^\intercal}^2
$$

$$
\mathbf{z} = \mathbf{d}\mathbf{1}_N^\intercal - \tfrac{1}{2}\mathbf{H}\mathbf{X}^\mathrm{f}, \tag{16}
$$

$$
\left\| \mathbf{y} - \mathbf{H}\mathcal{A}(\mathbf{x}^\mathrm{f}, \mathbf{y}, \boldsymbol{v}) \right\|_{\mathbf{R}^{-1}}^2 = \sum_{e=1}^{N} \left\| \mathbf{g}_{(\boldsymbol{v})}^{(e)} \right\|_{\mathbf{R}^{-1}}^2
$$

$$
\mathbf{g}_{(\boldsymbol{v})} = \left( \mathbf{I} - \mathbf{H}\mathbf{K}_{(\boldsymbol{v})} \right) \mathbf{d}\mathbf{1}_N^\intercal
$$

$$
- \mathbf{H}\mathbf{X}^\mathrm{f}
$$

$$
+ \tfrac{1}{2}\mathbf{H}\mathbf{K}_{(\boldsymbol{v})}\mathbf{H}\mathbf{X}^\mathrm{f},
$$

$$
\mathbf{K}_{(\boldsymbol{v})} = \mathbf{P}_{(\boldsymbol{v})}^\mathrm{f}\mathbf{H}^\intercal \mathbf{S}_{(\boldsymbol{v})}^{-1},
$$

$$
\mathbf{S}_{(\boldsymbol{v})} = \mathbf{H}\mathbf{P}_{(\boldsymbol{v})}^\mathrm{f}\mathbf{H}^\intercal + \mathbf{R}. \tag{17}
$$

Combining (14), (16), and (17) leads to the cost function form:

$$
\mathcal{J}(\boldsymbol{v}) = \sum_{e=1}^{N} \left[ \tfrac{1}{2} \left\| \mathbf{S}_{(\boldsymbol{v})}^{-1} \mathbf{z}^{(e)} \right\|_{\mathbf{H}\mathbf{P}_{(\boldsymbol{v})}^{\mathrm{f}} \mathbf{H}^\intercal}^2 + \tfrac{1}{2} \left\| \mathbf{g}_{(\boldsymbol{v})}^{(e)} \right\|_{\mathbf{R}^{-1}}^2 \right]
$$

$$
+ \sum_{j=1}^{g} \left( \beta^{(j)} v^{(j)} - \left( \alpha^{(j)} - 1 \right) \log \left( v^{(j)} \right) \right), \tag{18}
$$

which only requires the computation of the background covariance matrix $\mathbf{H}\mathbf{P}_{(\boldsymbol{v})}^\mathrm{f}\mathbf{H}^\intercal$ in observation space, thereby reducing the amount of computation and storage. The equivalent manipulations of the gradient lead to:

$$
\frac{\partial \mathcal{J}}{\partial v^{(j)}} = \sum_{e=1}^{N} \left[ \tfrac{1}{2} \left\| \mathbf{S}_{(\boldsymbol{v})}^{-1} \mathbf{z}^{(e)} \right\|_{\mathbf{H}\frac{\partial \mathbf{P}_{(\boldsymbol{v})}^{\mathrm{f}}}{\partial v^{(j)}}\mathbf{H}^\intercal}^2 \right.
$$

$$
- \mathbf{z}^\intercal \mathbf{H} \frac{\partial \mathbf{P}_{(\boldsymbol{v})}^\mathrm{f}}{\partial v^{(j)}} \mathbf{H}^\intercal \mathbf{S}_{(\boldsymbol{v})}^{-2} \mathbf{H}\mathbf{P}_{(\boldsymbol{v})}^\mathrm{f}\mathbf{H}^\intercal \mathbf{S}_{(\boldsymbol{v})}^{-1}\mathbf{z}
$$

$$
\left. + \mathbf{H}\frac{\partial \mathbf{K}_{(\boldsymbol{v})}}{\partial v^{(j)}} \left( \tfrac{1}{2}\mathbf{H}\mathbf{X}^\mathrm{f} - \mathbf{d}\mathbf{1}_N^\intercal \right) \right]
$$

$$
+ \beta^{(j)} - \left( \alpha^{(j)} - 1 \right) \frac{1}{v^{(j)}}, \tag{19}
$$

$$
\frac{\partial \mathbf{K}_{(\boldsymbol{v})}}{\partial v^{(j)}} = \frac{\partial \mathbf{P}_{(\boldsymbol{v})}^\mathrm{f}}{\partial v^{(j)}} \mathbf{H}^\intercal \mathbf{S}_{(\boldsymbol{v})}^{-1}
$$

$$
- \mathbf{P}_{(\boldsymbol{v})}^\mathrm{f}\mathbf{H}^\intercal \mathbf{H} \frac{\partial \mathbf{P}_{(\boldsymbol{v})}^\mathrm{f}}{\partial v^{(j)}} \mathbf{H}^\intercal \mathbf{S}_{(\boldsymbol{v})}^{-2}, \tag{20}
$$





where $\frac{\partial \mathbf{P}^{\mathrm{f}}_{(\boldsymbol{v})}}{\partial v^{(j)}} = \frac{\partial \boldsymbol{\rho}_{(\boldsymbol{v})}}{\partial v^{(j)}} \circ \mathbf{P}^{\mathrm{f}}$. Calculation of the gradient (19) requires computations only in observation space.

One will note that the form of our cost function (18) is similar to that of other hybrid approaches to data assimilation such as 3DEnVar (Hamill and Snyder, 2000).

## 3.2 Extension to multivariate localization functions

It is intuitively clear that different physical effects propagate spatially at different rates, leading to different correlation distances. Consequently, different state-space variables should be analyzed using different radii of influence. This raises the additional question of how to localize the covariance of two variables when each of them is characterized by a different radius of influence. One approach (Roh et al., 2015) is to use a multivariate compactly supported function (Askey, 1973; Porcu et al., 2013) to calculate the modified error statistics. We however wish to take advantage of already developed univariate compactly supported functions.

Petrie (Petrie, 2008) showed that true square-root filters such as the LETKF (Hunt et al., 2007) are not amenable to Schur product-based localization, therefore they need to rely on sequentially assimilating every observation space variable. Here we wish to combine both the ease-of-use of Schur product-based localization and the utility of multivariate localization techniques.

To this end, we introduce a commutative, idempotent, binary operation, $m : \mathbb{R}_{\geq 0} \times \mathbb{R}_{\geq 0} \to \mathbb{R}_{\geq 0}$, that computes a "mean localization function value" such as to harmonize the effects of different values of the localization radii. More explicitly, given $0 \leq a \leq b$, $m$ should have the properties that $m(a,a) = a$, $m(b,b) = b$, and $m(a,b) = m(b,a)$. We also impose the additional common sense property that $a \leq m(a,b) \leq b$. We consider here several simple mean functions $m$, as follows:

$$m_{\min}(\lambda_i, \lambda_j) = \min\{\lambda_i, \lambda_j\}, \tag{21a}$$

$$m_{\max}(\lambda_i, \lambda_j) = \max\{\lambda_i, \lambda_j\}, \tag{21b}$$

$$m_{\mathrm{mean}}(\lambda_i, \lambda_j) = (\lambda_i + \lambda_j)/2\,, \tag{21c}$$

$$m_{\mathrm{sqrt}}(\lambda_i, \lambda_j) = \sqrt{\lambda_i \lambda_j}, \tag{21d}$$

$$m_{\mathrm{rms}}(\lambda_i, \lambda_j) = \sqrt{\lambda_i^2 + \lambda_j^2}\Big/\sqrt{2}\,, \tag{21e}$$

$$m_{\mathrm{harm}}(\lambda_i, \lambda_j) = \frac{2\lambda_i \lambda_j}{\lambda_i + \lambda_j}. \tag{21f}$$

Assume that the variables $x_i$ and $x_j$ have the localization radii $r_i$ and $r_j$, respectively. We extend the definition of the localization matrix $\boldsymbol{\rho}$ to account for multiple localization radii associated with different variables as follows:

$$\boldsymbol{\rho} = \Big[ m\big(\ell(d(i,j)/r_i), \ell(d(j,i)/r_j)\big) \Big]_{1 \leq i,j \leq n} \tag{22}$$

The localization sub-matrix of state-space variables $x_i$ and $x_j$:

$$\begin{bmatrix} \ell(0) & \boldsymbol{\rho}_{i,j} \\ \boldsymbol{\rho}_{j,i} & \ell(0) \end{bmatrix} \tag{23}$$



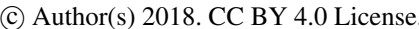



is a symmetric matrix for any mean function (21). When $\ell$ is a univariate compactly supported function, our approach implicitly defines its $m$-based multivariate counterpart.

One of the common criticisms of a distance assumption on the correlation of geophysical systems is that two variables in close proximity to each other might have very weak correlations. For example, in a model that takes into account the temperature and concentration of stationary cars at any given location, the two distinct types of information might not at all be correlated with each other. The physical distance between the two, however, is zero, and thus any single correlation function will take the value one and do not remove any spurious correlations. One can mitigate this problem by considering univariate localization functions for each pair of components $\ell_{i,j} = \ell_{j,i}$, and extending the localization matrix as follows:

$$\boldsymbol{\rho} = \Big[ m\big(\ell_{i,j}(d(i,j)/r_i), \ell_{j,i}(d(j,i)/r_j)\big) \Big]_{1 \leq i,j \leq n}. \tag{24}$$

We will not analyze further this extension in the paper.

### 3.3 Adaptive localization via a time-distributed cost function

We now seek to extend the 3D-Var-like cost function (14) to a time-dependent 4D-Var-like version. As the ensemble is essentially a reduced order model, we do not expect the accuracy of the forecast to hold over the long term as we might in traditional 4D-Var. We thereby propose a limited extension of the cost-function to include a small number $K$ of additional future times. Assuming that we are currently at the $i$-th time moment, the extended cost function reads:

$$
\begin{aligned}
\mathcal{J}_i(\boldsymbol{v}) = \sum_{e=1}^{N} \Bigg[ &\tfrac{1}{2} \Big\| \mathbf{S}_{(\boldsymbol{v}),i}^{-1} \mathbf{z}_i^{(e)} \Big\|_{\mathbf{H}\mathbf{P}_{(\boldsymbol{v}),i}^{\mathrm{f}}\mathbf{H}^\intercal}^2 \\
&+ \tfrac{1}{2} \Big\| \mathbf{g}_{(\boldsymbol{v}),i}^{(e)} \Big\|_{\mathbf{R}_i^{-1}}^2 \\
&+ \sum_{k=1}^{K} \tfrac{1}{2} \Big\| \mathbf{y}_{i+k} - \mathbf{H}\mathbf{x}_{(\boldsymbol{v}),i+k}^{f,(e)} \Big\|_{\mathbf{R}_{i+k}^{-1}}^2 \Bigg] \\
&+ \sum_{j=1}^{g} \Big( \beta_i^{(j)} v^{(j)} - \big(\alpha_i^{(j)} - 1\big) \log\big(v^{(j)}\big) \Big).
\end{aligned}
\tag{25}
$$

One will notice that in the 4D part of the cost function the future forecasts are now dependent on $\boldsymbol{v}$ as they are obtained by running the model from $\mathbf{x}_{i,(\boldsymbol{v})}^{\mathrm{a}}$. Similar to all other 4D ensemble approaches, this requires only the computation of the tangent linear model in order to derive the gradient; an adjoint model is not required. In fact all that is required is matrix vector products which can be approximated with finite differences (Tranquilli et al., 2017).

### 3.4 Algorithm

In practice, instead of dealing with the Gamma distribution parameters of $\alpha$ and $\beta$, we use the parameters $\bar{v}$ and $\mathrm{Var}(v)$, such that $\alpha = \frac{\bar{v}^2}{\mathrm{Var}(v)}$ and $\beta = \frac{\bar{v}}{\mathrm{Var}(v)}$. For the sake of simplicity we assume that all $\boldsymbol{v}$ are identically distributed, but this is not required for the algorithm to function. The initial guess for our minimization procedure is the vector of means. After minimizing the

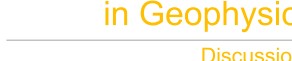
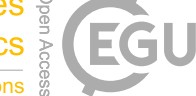
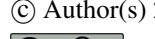

cost function, the radii for different components will be different. These radii, along with the corresponding localization and $m$ functions, are used to build the localization matrix $\rho$. An outline is presented in Algorithm 1.

---

**Algorithm 1** DEnKF Adaptive Localization Algorithm

---

**Require:** $[\bar{\upsilon}^{(j)}]_j, [\mathrm{Var}(\upsilon^{(j)})]_j$

1: $\alpha^{(j)} \leftarrow \dfrac{\bar{\upsilon}^{(j),2}}{\mathrm{Var}(\upsilon^{(j)})}$

2: $\beta^{(j)} \leftarrow \dfrac{\bar{\upsilon}^{(j)}}{\mathrm{Var}(\upsilon^{(j)})}$

3: $\boldsymbol{\upsilon}_0^* \leftarrow [\bar{\upsilon}^{(j)}]_j$

4: $\boldsymbol{\upsilon}^* \leftarrow \mathrm{argmin}_{\boldsymbol{\upsilon}} \mathcal{J}(\boldsymbol{\upsilon})$ {Eq (18)}

5: $\boldsymbol{r}^* \leftarrow \mathfrak{p}(\boldsymbol{\upsilon}^*)$ {Eq (10)}

6: $\boldsymbol{\rho} \leftarrow [m(l_i, l_j)]_{1 \leq i,j \leq n}$ {Eq (24)}

7: **with**

8: $\quad l_i \leftarrow \ell(d(i,j)/r^{(i),*})$ {Possibly Eq (8)}

9: $\quad l_j \leftarrow \ell(d(i,j)/r^{(j),*})$

10: **return** $\boldsymbol{\rho}$

---

## 4   Numerical Experiments and Results

In order to validate our methodology we carry out twin experiments under the assumption of identical perfect dynamical
systems for both the truth and the model. The analysis accuracy is measured by the spatio-temporally averaged root mean square error:

$$\mathrm{RMSE} = \sqrt{\frac{1}{n \cdot n_t} \sum_{i=1}^{n_t} \sum_{j=1}^{n} \left([\mathbf{x}_i^{\mathrm{t}}]_j - [\bar{\mathbf{x}}_i^{\mathrm{a}}]_j\right)^2}, \tag{26}$$

where $n_t$ is the number of data assimilation cycles (the number of analysis steps).

For each of the models we repeat the experiments with different values of the inflation constant $\alpha$ and report the best
(minimal RMSE) results.

### 4.1   Oracles

We will make use of oracles to empirically evaluate the performance of the multivariate approach to Schur product localization.
An oracle is an idealized procedure that produces close to optimal results by making use of all the available information, some
of which is unknown to the data assimilation system. In our case the oracle minimizes cost functions involving the true model
state. Specifically, in an ideal filtering scenario one seeks to minimize the error of the analysis with respect to the truth, i.e., the
cost function $\mathcal{J}(\mathbf{x}^{\mathrm{a}}) = \mathrm{RMSE}(\mathbf{x}^{\mathrm{a}} - \mathbf{x}^{\mathrm{t}})$. Our oracle will pick the best parameters, in this case the radii, that minimize the ideal
cost function $\mathcal{J}(\boldsymbol{\upsilon}) = \mathrm{RMSE}(\bar{\mathbf{x}}_{(\boldsymbol{\upsilon})}^{\mathrm{a}} - \mathbf{x}^{\mathrm{t}})$.



## 4.2 Lorenz'96

The 40 variable Lorenz model (Lorenz, 1996; Lorenz and Emanuel, 1998) is the first standard test employed. This problem is widely used in the testing of data assimilation algorithms.

### 4.2.1 Model Setup

5   The Lorenz'96 model equations:

$$\frac{dx_i}{dt} = (x_{i+1} - x_{i-2})x_{i-1} - x_i + F, \quad i = 1, \ldots n, \tag{27}$$

are obtained through a coarse discretization of a forced Burger's equation with Newtonian damping (Reich and Cotter, 2015). We impose $x_0 = x_n$, $x_{-1} = x_{n-1}$, and $x_{n+1} = x_1$, where $n = 40$. We take the canonical value for the external forcing factor, $F = 8$. Using known techniques for dynamical systems (Parker and Chua, 2012) one can calculate that this particular system has 13 positive Lyapunov exponents (Strogatz, 2014) and one zero exponent, with a fractal dimension of approximately 27.1.

The initial conditions used for experiments are obtained by starting with

$$[\mathbf{x}_0]_i = \begin{cases} 8 & i \neq 20 \\ 8.008 & i = 20 \end{cases}, \tag{28}$$

and integrating (27) forward for one time unit in order to reach the attractor.

The physical distance between $x_i$ and $x_j$ is the shortest cyclic distance between any two state variables:

15   $$d(i,j) = \min\{|i-j|, |n+i-j|, |n+j-i|\}, \tag{29}$$

where the distance between two neighbors is one.

For numerical experiments we consider a perfect model and noisy observations. We take a six hours assimilation window (corresponding to $\Delta t_{\text{obs}} = 0.05$ model time units), and calculate the RMSE on the assimilation cycle interval $[100, 1100]$ in the case of oracle testing, or $[5000, 55000]$ in the case of adaptive localization testing. Lorenz '96 is an ergodic system (Fatkullin and Vanden-Eijnden, 2004) and therefore its behavior in time for most initial conditions is the same as the behavior of all its possible phase spaces on any orbit around a strange attractor at any point in time, meaning that a long enough time averaged run should be the same as a collection of shorter space averaged runs. We use 10 ensemble members, and observe the thirty model variables $(2, 4, \ldots, 18, 20, 21, \ldots, 39, 40)$, with an observation error variance of one for each observed state.

### 4.2.2 Oracle Results

25   Figure 1 shows a visualization of multivariate oracle runs for the Lorenz'96 test problem using best constant radius, univariate oracle, and multivariate oracle utilizing the $m$-functions from (21). As one can see, the univariate oracle performs no better than a constant radius, while several of the multivariate approaches provide much better results. This indicates that the problem is better suited for multivariate localization. The worst $m$-function results are given by $m_{\min}$. The other functions perform



similarly, and in further experiments we will only consider $m_{\text{mean}}$ as a representative and easy to implement option. We note that for Lorenz-96 with DEnKF and our experimental setup, no 3D univariate adaptive localization scheme beats the best constant localization radius.

We also test both the validity of arbitrarily grouping the radii and the validity of using a time-distributed cost function ((25)).
Figure 2 presents results for arbitrary radii groupings and for a limited run 4D approach. There is significant benefit in using more radii groupings, but marginal benefit from the 4D approach.

### 4.2.3 Adaptive Localization Results

Adaptive localization results for Lorenz'96 are shown in figure 3. As accurately predicted by the oracle results, an adaptive approach for this model cannot do better than the best univariate radius (figure 1).

### 4.3 Multivariate Lorenz'96

The canonical Lorenz'96 model is ill suited for multivariate adaptive localization as each variable in the problem behaves identically too all the others. This means that for any univariate localization scheme a constant radius is close to optimal.

### 4.3.1 Model Setup

We modify the problem in such a way that the average behavior remains very similar to that of the original model, but that instantaneous behavior requires different localization radii. In order to accomplish this we use the time-dependent forcing function that is different for each variable:

$$[\mathbf{F}(t)]_i = 8 + 4\cos\left(\omega\Big(t + \frac{(i-1)\mod q}{q}\Big)\right), \tag{30}$$

where $\omega = 2\pi$ (in the context of Lorenz'96 the equivalent period is 5 days), $i$ is the variable index, and $q$ is an integer factor of $n$. Here we set set $q = 4$.

For each individual variable the forcing value cycles between 4 and 12, with an average value of 8, just like in the canonical Lorenz'96 formulation. If taken to be a constant, the forcing factor of 4 will make the equation lightly chaotic with only one positive Lyapunov exponent, whilst a constant value of 12 will make the dynamics to have about 15 positive Lyapunov exponents. Our modified system still has the same average behavior with 13 positive Lyapunov exponents. The mean doubling times of the two problems are also extremely similar at around $0.36$. This is the ideal desired behavior. Figure 4 shows a comparison of numerically computed covariance matrices for this modified problem. This showcases that an adaptive approach to the instantaneous covariance is required.

Data assimilation will be carried out over the interval $[500, 5500]$. The rest of the problem setup is identical to that of the canonical Lorenz'96.





### 4.3.2 Adaptive Localization Results

Figure 5 shows results with the Multivariate Lorenz'96 model, with $g = 4$ radii groups, and the 4D parameter set to $K = 1$. We note that the filter spin-uptakes significantly longer for the adaptive localization case than for the constant univariate case. Consequently, the assimilation cycles are chosen in the time interval $[500, 5500]$ units. An idea to mitigate this might be to run

the filter with a constant radius for a few assimilation cycles before switching to the adaptive localization strategy, such as to allow the filter to quickly catch up with the shadow attractor trajectory.

Tightly coupled models like the multivariate Lorenz'96 have rapidly diverging solutions, and constraining them requires more information about the underlying dynamics. Incorporating future observations and adding degrees of freedom to the cost function increase the performance of our analysis. In the limiting case of one radius per variable and general information from

the future one approaches a variant of 4DenVar, which is in principle superior to any pure filtering method.

### 4.4 Quasi-geostrophic model

The 1.5 layer quasi-geostrophic model of Sakov and Oke (Sakov and Oke, 2008), obtained by non-dimensionalizing potential vorticity, is given by the equations:

$$q_t = -\psi_x - \varepsilon J(\psi, q) - A \triangle^3 \psi + 2\pi \sin(2\pi y),$$

$$\triangle \psi - F\psi = q, \quad J(\psi, q) \equiv \psi_x q_y - \psi_y q_x. \tag{31}$$

The variable $\psi$ can be thought of as the stream function, and the spatial domain is the square $(x, y) \in [0, 1]^2$. The constants are $F = 1600$, $\varepsilon = 10^{-5}$, and $A = 2 \times 10^{-11}$. We use homogeneous Dirichlet boundary conditions.

A second order central finite difference spatial discretization of the Laplacian operator $\triangle$ is performed over the interior of a $129 \times 129$ grid, leading to a model dimension $n = 127^2 = 16,129$. The time discretization is the canonical fourth-order explicit

Runge-Kutta method with a timestep of 1 time units. The Helmholtz equation on the right-hand side of (31) is solved by an offline pivoted sparse Cholesky decomposition. $J$ is calculated via the canonical Arakawa approximation (Arakawa, 1966; Ferguson, 2008). The $\triangle^3$ operator is implemented by repeated application of our discrete Laplacian.

The time between consecutive observations is 5 time units, and the model is run for 3300 such cycles. The first 300 cycles, corresponding to the filter spin-up, are discarded, therefore the assimilation interval is $[300, 3300]$ time units. Observations are

performed with a standard 300 component observation operator, as shown in Figure 6. An observation error variance of 4 is taken for each component. The physical distance between two component is defined as:

$$d(i, j) = \sqrt{(i_x - j_x)^2 + (i_y - j_y)^2}, \tag{32}$$

with $(i_x, i_y)$ and $(j_x, j_y)$ are the spatial grid coordinates of the state space variables $i$ and $j$, respectively.

The number of positive Lyapunov exponents of this model is currently unknown, thus we will take a conservative 25 ensem-

ble members whose initial states are derived from a random sampling of a long run of the model. A typical model state along with the observation points is given in Figure 6.



This model has been tested extensively with both the DEnKF and with various localization techniques (Sakov and Bertino, 2011; Bergemann and Reich, 2010; Moosavi et al., 2018).

### 4.4.1 Adaptive Localization Results

The adaptive localization results for the Quasi-geostrophic problem are shown in Figure 7. One can readily see that for certain values of the inflation factor the adaptive localization procedure results in significant reductions in analysis error, while for other values no significant benefits are observed.

The empirical utility of the adaptive localization technique is further analyzed in Figure 8 which compares the error results of a suboptimal constant radius with that of an adaptive run with the mean parameter set to the same values as the constant ones. The adaptive results are—except in a few cases of filter divergence—always as good or better than their constant localization counterparts. This indicates that our localization scheme is significantly better than a corresponding sub-optimal constant scheme with the same parameters, as is typically the case in real-world production codes. This opens up the possibility of adapting existing systems that use a conservative suboptimal constant localization to an adaptive localization scheme.

Figure 9 shows a sample of the radii obtained by the adaptive algorithm. The initial period of algorithm spin-up is clearly visible. One notes that the adaptive scheme is can discern—on average—whether the observations or the model are to be trusted more.

## 5 Conclusions

This paper proposes a novel Bayesian approach to adaptive Schur product-based localization. A multivariate approach is developed, where multiple radii corresponding to different types of variables are taken into account. The Bayesian approach is solved by constructing 3D and 4D cost functions, and minimizing them to obtain the maximum aposteriori estimates of the localization radii. We show that in the case of the DEnKF these cost functions and their gradients are computationally inexpensive to evaluate, and can be relatively easily implemented within existing frameworks. We provide a new approach for assessing the performance of adaptive localization approaches through the use of restricted cost function oracles.

The adaptive localization approach is tested using the Lorenz'96 and the Quasi-Geostrophic models. Somewhat surprisingly, the adaptivity produces better results for the larger Quasi-Geostrophic problem. This may be due to the ensemble analysis anomaly independence assumption made in Sect. 3.1, an assumption that holds better for a large system with sparse observations than for a small tightly coupled system with dense observations. The performance of the adaptive approach on the small, coupled Lorenz'96 system is increased by using multivariate and 4D extensions of the cost function.

We believe that the algorithm presented herein have a strong potential to improve existing geophysical data assimilation systems that use ensemble based filters such as the DEnKF. In order to avoid filter divergence in the long term, these systems often use a conservative localization radius and a liberal inflation factor. The QG model results indicate that, in such cases, our adaptive method outperforms the approach based on a constant localization. The new adaptive methodology can replace the existing approach with a relatively modest implementation effort.



Future work to extend the methodology includes finding good approximations of the probability distribution of the localization parameters, perhaps through a Machine Learning approach, and reducing the need of assumption that the ensemble members are independent identically distributed random variables.

*Author contributions.* All authors contributed equally to this work

5 *Competing interests.* The authors declare that they have no conflict of interest.

*Acknowledgements.* This work was supported by awards AFOSR DDDAS FA9550–17–1–0015, NSF CCF–1613905, NSF ACI–17097276, and by the Computational Science Laboratory at Virginia Tech.



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

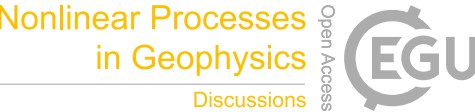



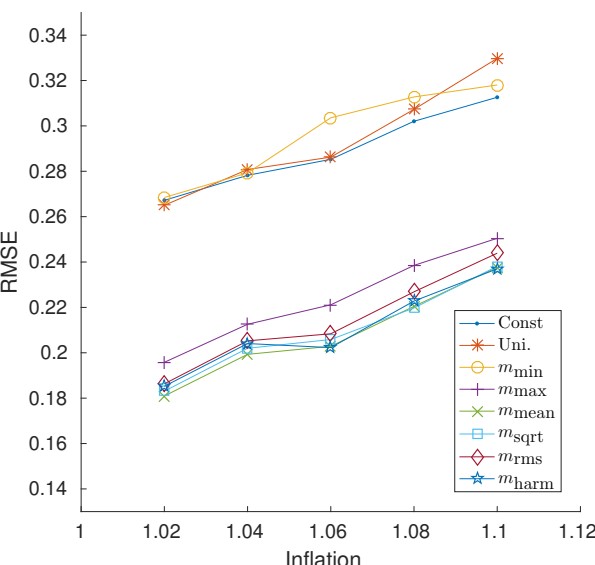

**Figure 1.** GENERALISED MEAN ORACLES FOR LORENZ'96. Comparison of the various $m$-based multivariate localization techniques with that of standard univariate localization. Results obtained using the Lorenz'96 model over the assimilation cycles $[100, 1100]$. The particular localization scheme used is an adaptive oracle search that minimizes the error of the Schur-product localized analysis with respect to the truth. Each of the forty of variables is given an independent radius in the multivariate case. The schemes (21) that closely mirror an unbiased mean, namely $m_{\text{mean}}$, $m_{\text{sqrt}}$, $m_{\text{rms}}$, and $m_{\text{harm}}$, yield the best results, while the conservative scheme $m_{\text{min}}$ performs no better than a univariate approach.





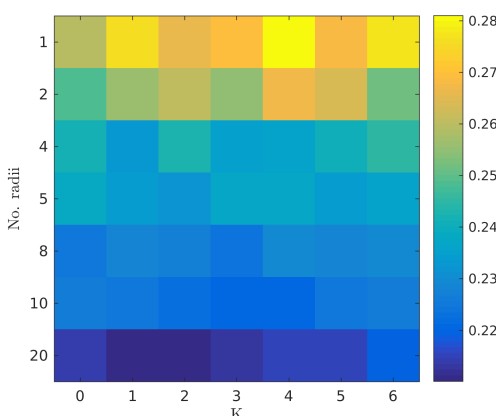

**Figure 2.** 4D ORACLE FOR LORENZ'96. Plot of the RMSE for a radius oracle. The $y$-axis represents the number of independent radii values (groups of the model state components). The assimilation cycle interval is $[100, 1100]$, a fixed inflation value of $\alpha = 1.02$ and the function $m_{\text{mean}}$ are used. As can be seen, there is significant benefit in using more radii groupings, but marginal benefit from the 4D approach.



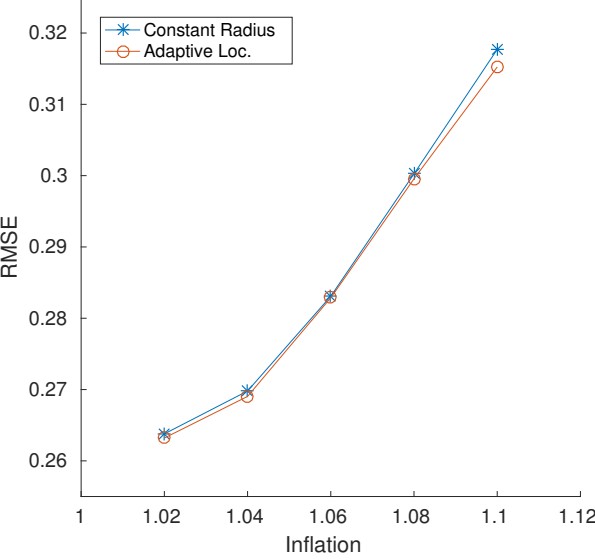

**Figure 3.** LORENZ'96 ADAPTIVE LOCALIZATION RESULTS Comparison of the best univariate localization radius results with their corresponding adaptive localization counter parts. $\bar{\upsilon}$ was varied by additive factors of one of $-1, -0.5, +0, +0.5, +1$ with respect to the best univariate radius with $\mathrm{Var}(\upsilon)$ chosen to be one of $1/8, 1/4, 1/2, 1, 2$. As was predicted, no adaptive scheme that takes into account only currently available information can do well for Lorenz'96.




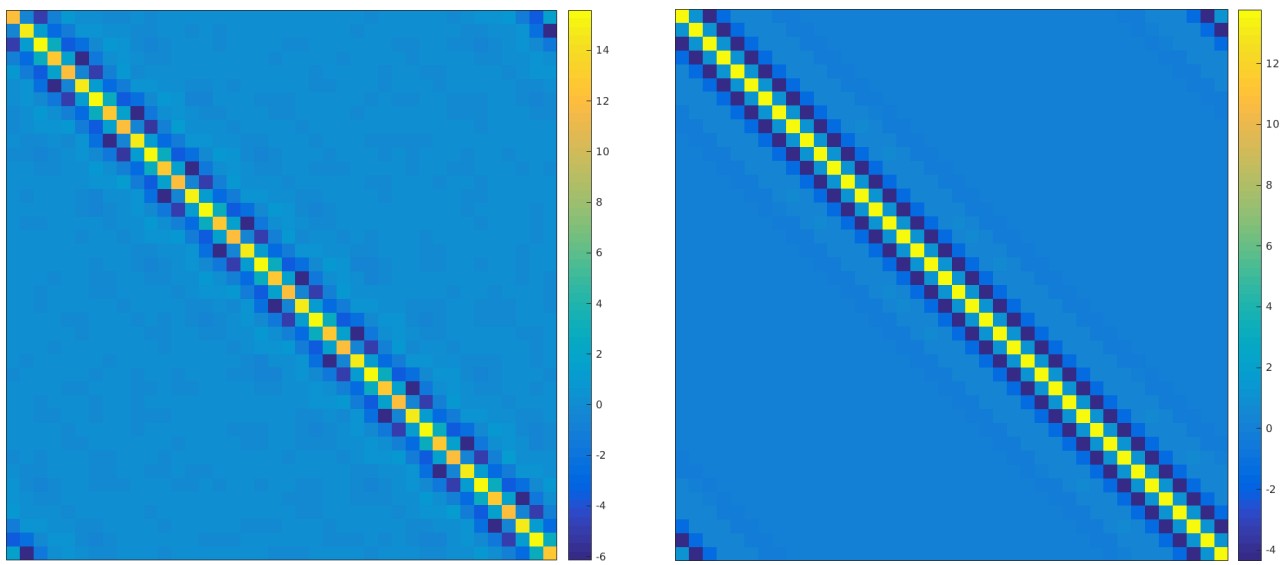

**Figure 4.** Comparison of ensemble covariance matrices for the multivariate Lorenz'96 equations for a single time-step (left) with that for time-averaged run (right) for a large number of ensemble members. There is considerable difference between the size of non-diagonal entries for different state variables over the course of a single step, but this difference disappears after averaging. This indicates that for this problem the best constant univariate radius is the same as for the canonical Lorenz'96 model, but that instantaneous adaptive radii are different.





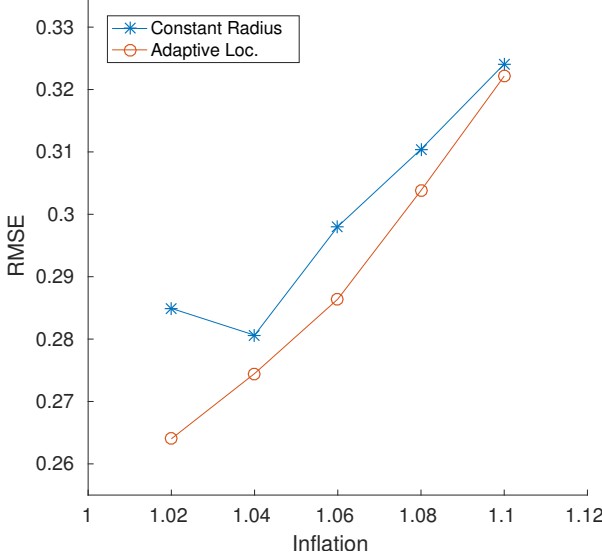

**Figure 5.** MULTIVARIATE LORENZ'96 4D ADAPTIVE LOCALIZATION. A Gauss localization function is employed. The inflation is constant, and we test values $\alpha$ from 1.02 to 1.1, represented on the x-axis. The localization radius is varied in increments of $0.5$ over the range $[0.5, 16]$ for the constant case. The minimal errors are shown in the graph, and the corresponding radii are used as mean inputs into our adaptive algorithm. For the adaptive case we choose four arbitrary groupings of radii ($g = 4$) with the mean function $m_{\text{mean}}$. The parameters $\bar{v}$ take one of three possible values, the optimal constant radius $-1$, $+0$, and $+1$. The parameters $\text{Var}(v)$ take one value from $\{1/4, 1, 4\}$. The 4D variable is set to $K = 1$ to look at an additional observation in the future.





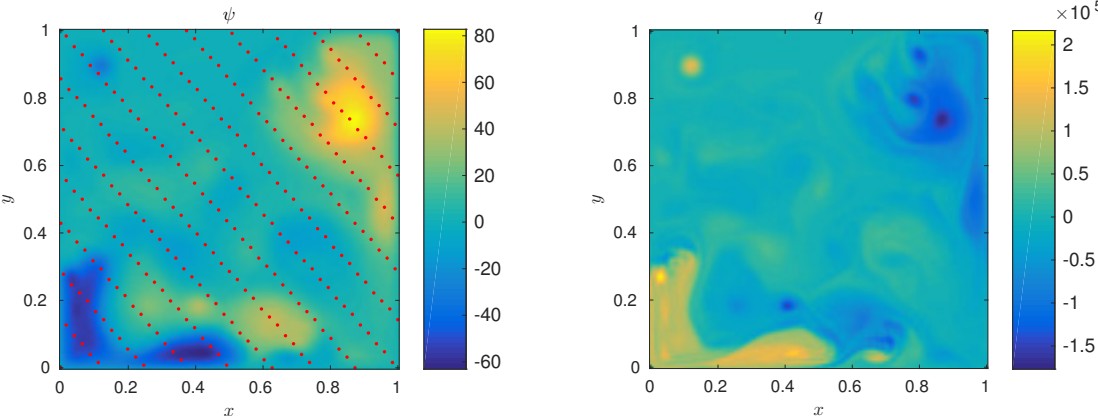

**Figure 6.** QUASI-GEOSTROPHIC MODEL. A typical model state of the 1.5 layer Quasi-Geostrophic model. The left panel shows the stream function values, with the red dots representing the locations of the variables that are observed. The right panel shows the corresponding vorticity.




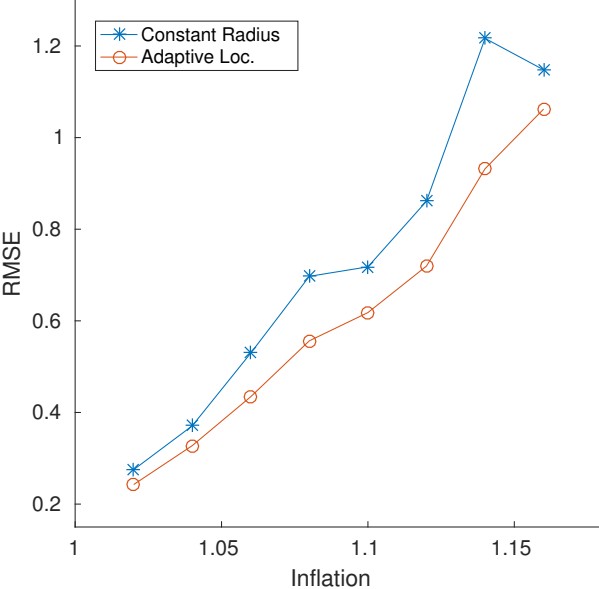

**Figure 7.** QUASI-GEOSTRPHIC MODEL ADAPTIVE LOCALIZATION. A Gauss localization function is used. The inflation factor is kept constant, and we test $\alpha$ values from 1.02 to 1.16 (represented on the x-axis). The constant localization radius varies in increments of 5 over the range $[5, 45]$. Only the best results are plotted, and are used as the mean seeds for the adaptive algorithm. For the adaptive case we vary the parameters $\bar{v}$ to by taking one of three possible values, differing from the optimal constant radius by $-5$, $0$, and $+5$. The adaptive $\mathrm{Var}(v)$ takes one of the values $1/2$, $1$, $2$, and $4$. As can be seen, for higher values of inflation, the adaptive localization approach performs stellarly.

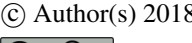


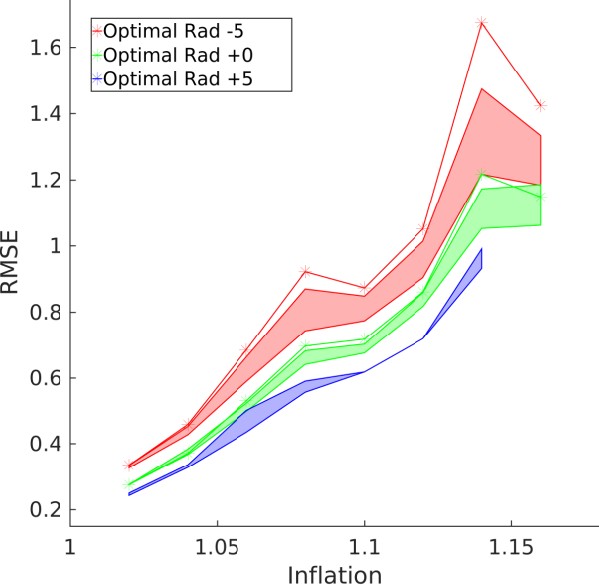

**Figure 8.** QUASI-GEOSTRPHIC MODEL ADAPTIVE LOCALIZATION RAW RMSE. A better representation of how well the adaptive local-ization scheme works is by showing its consistency. The green line represents the same optimal constant localization radius as in Figure 7. The red line represents the error for the radius -5 units below, and the blue line, if it had not experienced filter divergence would have repre-sented +5. The correspondingly colored areas represent the ranges of error of the adaptive localization scheme obtained by fixing the mean, $\bar{v}$, to be that of the constant scheme and ranging over the variance. As can be seen, the adaptive scheme generally outperformed the constant scheme for almost all ranges of the variance and even managed to not suffer from filter divergence for some values of the variance in the +5 case, thereby providing evidence for the fact that an overestimation of the variance coupled with a useful estimate of the mean, might produce more useful results than the corresponding constant counterpart.



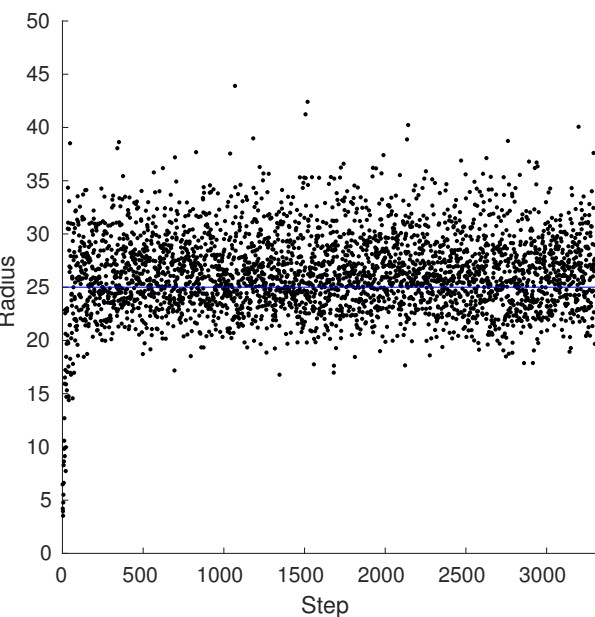

**Figure 9.** QGSO SAMPLE RADII. For the configuration in Figure 7 with $\alpha = 1.08$, $\bar{v} = 25$, and $\mathrm{Var}(v) = 4$. Each dot represents a radius, with the line representing the mean. One will notice that during the spin-up time the algorithm is much more conservative with the radius of influence signaling that there should be an over-reliance on the observations instead of the model prior. After the spin-up time however, the algorithm tends to select radii greater than the mean, signifying a greater confidence in the observations.