# Peer review of "A Bayesian Approach to Multivariate Adaptive Localization in Ensemble-Based Data Assimilation with Time-Dependent Extensions"

_Nonlinear Processes in Geophysics, 2018_

## Referee Comment (RC1) · Anonymous Referee #1 · 6 Dec 2018

Iteration: Initial submission

General Comments

This study presents a new method whereby a Bayesian problem is constructed for an adaptive Schur-product-based covariance localization method in ensemble Kalman filtering. There are some novel ideas that have been introduced, like the construction of Bayesian three and four-dimensional cost functions and their subsequent minimization to solve for "maximum likelihood" localization radii. The mathematical methodology in

itself is straight forward for the data assimilation expert reader, but some assumptions and concepts need to be further explained to reach a broader audience. The topic of the manuscript is within the scope of this journal. However, the manuscript contains limitations that need to be addressed before it can be considered for final publication. Specific details are provided in the following sections. Therefore, my recommendation is that major revisions are required.

Specific Comments

1. The manuscript requires some rearrangement. It is unclear as to why the introduction began by describing a model. This part is more suited for the "Background" section. 2. The motivation, objectives, and methods of the study should be clearly stated in the introduction. Some of these are actually discussed in the "Conclusions." Please make appropriate revisions. 3. Expand your literature review as to include other studies that have done applications of adaptive localization in ensemble methods 4. P5: the uncertainty in space is represented by pi(x | y, v), but according to equation (12), it should read pi(x | y,v) 5. P6: for equation (15), you may want to add that the chain rule was also used, in case you want to reach out to students, as derivations might not be too trivial 6. P7: equation (18) takes the form of an EnVar-like cost function. Comment on the applicability of your univariate or multivariate adaptive localization technique for hybrid EnVar data assimilation systems and also discuss limitations 7. P9, L3-10: Clarify if univariate localization functions were used and whether the extension of the localization matrix was done in the present study. Would this help in making your methodology suitable for testing with more complex geophysical models? 8. P10, L11-17: How do oracles compare to optimal parameter estimation as employed in control theory, with many applications in variational data assimilation? 9. P11, L22: Comment on the choice of 10 members for the ensemble and clarify if varying this number can impact your results 10. There are some mathematical notation typos, please revise all equations thoroughly, and make sure to conform to the mathematical notation standards of NPG for all scalars, vectors, matrices, etc. 11. Given the results with a simple geophysical model, provide a brief overview on how this methodology could be tested with a complex numerical weather prediction model (e.g. regional/convection allowing models)? 12. Provide more quantitative estimates in the "Results" and in the "Conclusions" sections

Technical Corrections

Typos

1. P1. L2-3: change to "variables," remove the period, and use lowercase for "Recognizing," insert a period after "property," and capitalize "adaptive" 2. P2, L5: revise the "in-text" citation, it should be: Le Dimet and Talagrand, 1986. 3. P2, L21: change to: the analysis cycles in space 4. P3, L2: add a comma before "such as" 5. P3, L19: remove "respectively" and insert it after P=... and preceded by a comma 6. P4, L1: add a comma before "which" 7. P5, L6: Insert a space before ".In," add a comma after "paper," and change to "Here, we consider" 8. P9, L20-21: change to "all that is required are," and add a comma before which 9. P10, L2: remove comma before "are" 10. P11, L17: change to: "For the numerical" 11. P14, L14: remove "is" before "can discern" 12. P14, L28: replace "have" with "has"

Figures

It is nice to see how descriptive all figure captions are (which is usually not the case with most research articles). However, corresponding "in-text" descriptions are lacking, try to incorporate some of these explanation on the results section.

---

## Referee Comment (RC2) · Anonymous Referee #2 · 18 Dec 2018

**1   General comments**

The manuscript presents a new approach to adaptive localization of ensemble background error covariance for ensemble data assimilation. Localization parameters are estimated by minimizing a variational DA-like cost function with additional terms for localization parameters. The authors extend the problem to support multiple localization radii for different state variables. The paper contains new and significant results. I recommend revising the manuscript to address several issues, please see specific

comments below.

**2 Specific comments**

- Several important references are missing from the Introduction, e.g. Menetrier et al, 2015 (https://journals.ametsoc.org/doi/10.1175/MWR-D-14-00157.1), Flowerdew, 2015 (https://www.tandfonline.com/doi/full/10.3402/tellusa.v67.25257) considered optimal localization, Buehner et al., 2015 (https://www.tandfonline.com/doi/full/10.3402/tellusa.v67.28027) suggested scale-dependent localization.

- I suggest rearranging the text so that everything related to multivariate localization is in section 3.2 (Extension to multivariate localization functions). I think describing univariate case first, and then introducing groups for different localization radii (currently P5, L7-13) when extending to multivariate localization functions might improve the manuscript readability.

- Questions and comments on the experiments

  - There seems to be a contradiction between 4.2 and 4.3. P11, L27-28 state that the problem is better suited for multivariate localization, while P12, L11-12 state that the canonical L96 model is ill suited for multivariate adaptive localization.

  - I would like to see more details on the L96 multivariate localization experiment setup (Figure 5). Were the groups fixed throughout the experiment? How were they chosen? Did the groups use the same mean and variance parameters at each assimilation cycle? If the groups were fixed, it would be interesting to see how the estimated localization radii vary for different groups throughout the experiment.

- I see that in L96 experiments half of the domain is more sparsely observed than the other. Introduction (P2, L34-35) states that optimal localization may depend on observation properties. In L96 experiments, did you see evidence of the optimal localization radii being dependent on observation density?
- Section 4.4.1, Figure 8. If instead of fixing the mean parameter to be the same as the constant suboptimal radius at each assimilation cycle, the adaptive localization radius estimated at the previous DA cycle was used to estimate the mean parameter, would the adaptive localization radius converge to the optimal one after some DA cycling?

- Questions on extensions of the method

  I think it would be good to address the following questions in the manuscript to show more clearly how the method may fit into existing large geophysical data assimilation applications.

  - How does this method extend to the ensemble DA algorithms other than DEnKF?
  - For large DA applications like NWP, ensemble filters similar to DEnKF typically assimilate observations sequentially, and use $P^f H^T$ localization instead of Schur-product $P^f$ localization which becomes too expensive. Would the method still be applicable in this case, and if so, how would it change?
  - Do you have a recommendation on how the groups for multivariate adaptive localization should be chosen?

**3 Technical comments**

What is subscript $i$ in Equation 10?

---

## Author Response (AR1)

**I. ADDITIONS**

We have added a citation to our own problems package that was used in the creation of this paper. We have also added rough results about the amount of Lyapunov exponents and fractal dimension of the QGSO model.

Some additional citations other than the ones asked for by the referees have been added in order to paint a more complete picture of the current state of knowledge.

A sentence has been added thanking the referees for their input into the quality of the paper.

Oracles have been clarified a bit and the introduction has been expanded.

Some duplicate information has also been removed.

**II. RESPONSE TO REFEREE 1**

We thank the referee for the many technical suggestions.

**A. Specific Comments**

*1. The manuscript requires some rearrangement. It is unclear as to why the introduc- tion began by describing a model. This part is more suited for the Background section.*

This makes sense. Done.

*2. The motivation, objectives, and methods of the study should be clearly stated in the introduction. Some of these are actually discussed in the Conclusions. Please make appropriate revisions.*

The introduction has been fleshed out to include more of our motivations and aims.

*3. Expand your literature review as to include other studies that have done applications of adaptive localization in ensemble methods*

This has been done, in conjunction to the recommendations of referee 2.

*4. P5: the uncertainty in space is represented by $pi(x|y, v)$, but according to equation (12), it should read $pi(x|y, v)$*

We are not sure what this comment is referring to.

*5. P6: for equation (15), you may want to add that the chain rule was also used, in case you want to reach out to students, as derivations might not be too trivial*

This is reasonable, and has been done. An additional comment about symmetric semi-positive definite matrices was also added.

*6. P7: equation (18) takes the form of an EnVar-like cost function. Comment on the applicability of your univariate or multivariate adaptive localization technique for hy- brid EnVar data assimilation systems and also discuss limitations*

An additional comment has been made discussing this.

*7. P9, L3-10: Clarify if univariate localization functions were used and whether the extension of the localiza- tion matrix was done in the present study. Would this help in making your methodology suitable for testing with more complex geophysical models?*

We have clarified that we did not use multiple localization functions in this paper, removing doubt that multivariate localization was used. We are not sure how this point related to more complex geophysical models, however we have clarified that point separately.

*8. P10, L11-17: How do oracles compare to optimal parameter estimation as employed in control theory, with many applications in variational data assimilation?*

A comment was added clarifying that oracles are a type of minimal variance estimator of a full space in a restricted space.

> *9. P11, L22: Comment on the choice of 10 members for the ensemble and clarify if varying this number can impact your results*

A comment was added clarifying the choice of 10 ensemble members in Lorenz '96. The question of varying the amount of ensemble members has been explored in other literature, and is outside the scope of this paper.

> *10. There are some mathematical notation typos, please revise all equa- tions thoroughly, and make sure to conform to the mathematical notation standards of NPG for all scalars, vectors, matrices, etc.*

All mathematical notation was changed to conform with the NPG guidelines, which mainly meant changing vectors to be boldface italics. The only Exception being the localization matrix $\rho$, as we could not find a way to have non-italic greek letters in conventional latex. Some other notation has been clarified (like indexing into columns using MATLAB syntax) in order to improve readability.

> *11. Given the results with a simple geophysical model, provide a brief overview on how this methodology could be tested with a complex numerical weather prediction model (e.g. regional/convection allowing models)?*

A sentence was added in the conclusion about the possibility of applying this approach to the WRF model.

> *12. Provide more quantitative estimates in the "Results" and in the "Conclusions" sections*

Quantitative estimates were previously provided in the figures. The figure captions have not been changed, but discussion about the results has also been provided where the figures are cited, with some additional quantitative estimates.

**B.   Technical Corrections**

All relevant typos have been addressed. Some comments in the typo section would have erroneously changed the semantics of several key statements and have thus been ignored.

More detailed information was added to the places where the figures are referenced, though we feel full duplication of the information would not be a good presentation of the information in the final publication.

**III.   RESPONSE TO REFEREE 2**

We thank the referee for many detailed suggestions and interesting questions.

**A.   Specific comments**

> *Several important references are missing from the Introduction, e.g. Menetrier et al, 2015, Flowerdew, 2015 considered optimal localization, Buehner et al., 2015 suggested scale-dependent localization.*

These have been added.

> *I suggest rearranging the text so that everything related to multivariate localization is in section 3.2 (Extension to multivariate localization functions). I think describ- ing univariate case first, and then introducing groups for different localization radii (currently P5, L7-13) when extending to multivariate localization functions might improve the manuscript readability.*

This is a good point. In order to make the transition smoother and emphasize the fact that the multivariate approach is important, it has been moved into its own section in front of the Bayesian Approach section.

> *There seems to be a contradiction between 4.2 and 4.3. P11, L27-28 state that the problem is better suited for multivariate localization, while P12, L11- 12 state that the canonical L96 model is ill suited for multivariate adaptive localization.*

There is no contradiction between 4.2 and 4.3. Although it is a bit confusing, Lorenz '96 was only tested with univariate adaptive localization, and the prediction was that the problem is not suited for this. This has been clarified in (the previous) 4.2.3 by adding the word 'univariate'.

> *I would like to see more details on the L96 multivariate localization experi- ment setup (Figure 5). Were the groups fixed throughout the experiment? How were they chosen? Did the groups use the same mean and variance parameters at each assimilation cycle? If the groups were fixed, it would be interesting to see how the estimated localization radii vary for different groups throughout the experiment.*

This segues into the next point about the multivariate Lorenz '96. Whereby in the normal Lorenz '96 there is no sensible way to create groupings (other than each variable being in its own group), We essentially create artificial groups through the varying forcing. All this has been clarified in the text.

> *I see that in L96 experiments half of the domain is more sparsely observed than the other. Introduction (P2, L34-35) states that optimal localization may depend on observation properties. In L96 experiments, did you see evidence of the optimal localization radii being dependent on observation density?*

An graph of the multi-group localization radii has been generated and discussed in the text, showcasing the differences between the radii of sparsely and fully observed groups.

> *Section 4.4.1, Figure 8. If instead of fixing the mean parameter to be the same as the constant suboptimal radius at each assimilation cycle, the adaptive localization radius estimated at the previous DA cycle was used to estimate the mean parameter, would the adaptive localization radius con- verge to the optimal one after some DA cycling?*

The idea of doing an online radius mean estimate has been explored, but unfortunately as the optimal radii between steps are weakly correlated, and are not normally distributed, this often lead to suboptimal convergence of the radius estimate, and often lead to filter divergence. This idea is outside the scope of this paper in our opinion.

**B.   Questions on Extensions**

How does this method extend to the ensemble DA algorithms other than DEnKF?

A paragraph has been added discussing this in the 'Solving the Optimization Problem' section, as the answer is closely tied with that problem.

> For large DA applications like NWP, ensemble filters similar to DEnKF typically assimilate observations sequentially, and use $P^f H^T$ localization instead of Schur-product $P^f$ localization which becomes too expensive. Would the method still be applicable in this case, and if so, how would it change?

This is already done in the paper! This has been clarified in the section discussing this approach.

> Do you have a recommendation on how the groups for multivariate adaptive localization should be chosen?

A paragraph discussing this has been added to the multivariate Lorenz '96 localization results.

**C.   Technical comments**

What is subscript $i$ in Equation 10?

The subscript $i$ has been removed from equation (10). It was a typo holdover from a previous version of the manuscript.

[revised manuscript text omitted]

$$m_{\min}(\lambda_i, \lambda_j) \quad = \min\{\lambda_i, \lambda_j\},$$

$$m_{\max}(\lambda_i, \lambda_j) \quad = \max\{\lambda_i, \lambda_j\},$$

25  $$m_{\mathrm{mean}}(\lambda_i, \lambda_j) \quad = (\lambda_i + \lambda_j)/2\,,$$

$$m_{\mathrm{sqrt}}(\lambda_i, \lambda_j) \quad = \sqrt{\lambda_i \lambda_j}\,,$$

$$m_{\mathrm{rms}}(\lambda_i, \lambda_j) \quad = \sqrt{\lambda_i^2 + \lambda_j^2}\big/\sqrt{2}\,,$$

$$m_{\mathrm{harm}}(\lambda_i, \lambda_j) \quad = \frac{2\lambda_i \lambda_j}{\lambda_i + \lambda_j}.$$

 Extending this idea to true a square-root filter, like the ETKF, would require significant algebraic
30  manipulation,

account for multiple localization radii associated with different variables as follows:

$$\boldsymbol{\rho} = \Big[ m\big(\ell(d(i,j)/r_i), \ell(d(j,i)/r_j)\big) \Big]_{1 \le i,j \le n}$$

The localization sub-matrix of state-space variables $x_i$ and $x_j$:

$$\begin{bmatrix} \ell(0) & \boldsymbol{\rho}_{i,j} \\ \boldsymbol{\rho}_{j,i} & \ell(0) \end{bmatrix}$$

is a symmetric matrix for any mean function . When $\ell$ is a univariate compactly supported function, our approach implicitly defines its $m$-based multivariate counterpart.

One of the common criticisms of a distance assumption on the correlation of geophysical systems is that two variables in close proximity to each other might have very weak correlations. For example, in a model that takes into account the temperature and concentration of stationary cars at any given location, the two distinct types of information might not at all be correlated with each other. The physical distance between the two, however, is zero, and thus any single correlation function will take the value one and do not remove any spurious correlations. One can mitigate this problem by considering univariate localization functions for each pair of components $\ell_{i,j} = \ell_{j,i}$, and extending the localization matrix as follows:

$$\boldsymbol{\rho} = \Big[ m\big(\ell_{i,j}(d(i,j)/r_i), \ell_{j,i}(d(j,i)/r_j)\big) \Big]_{1 \le i,j \le n}.$$

We will not analyze further this extension in the heuristics which are outside the scope of this paper.

**4.2 Adaptive localization via a time-distributed cost function**

We now seek to extend the 3D-Var-like cost function (18) to a time-dependent 4D-Var-like version. As the ensemble is essentially a reduced order model, we do not expect the accuracy of the forecast to hold over the long term as we might in traditional 4D-Var. We thereby propose a limited extension of the cost-function to include a small number $K$ of additional future times. Assuming that we are currently at the $i$-th time moment, the extended cost function reads:

$$\mathcal{J}_i(\boldsymbol{v}) = \sum_{e=1}^{N} \left[ \tfrac{1}{2} \left\| \mathbf{S}_{(\boldsymbol{v}),i}^{-1} \mathbf{z}_i^{(:,e)} \right\|_{\mathbf{H}\mathbf{P}_{(\boldsymbol{v}),i}^{\mathrm{f}} \mathbf{H}^{\mathsf{T}}}^2 \right.$$
$$+ \tfrac{1}{2} \left\| \mathbf{g}_{(\boldsymbol{v}),i}^{(:,e)} \right\|_{\mathbf{R}_i^{-1}}^2$$
$$\left. + \sum_{k=1}^{K} \tfrac{1}{2} \left\| \boldsymbol{y}_{i+k} - \mathbf{H}\mathbf{x}_{(\boldsymbol{v}),i+k}^{f,(:,e)} \right\|_{\mathbf{R}_{i+k}^{-1}}^2 \right]$$
$$+ \sum_{j=1}^{g} \left( \beta_i^{(j)} \upsilon^{(j)} - \left( \alpha_i^{(j)} - 1 \right) \log \left( \upsilon^{(j)} \right) \right). \tag{25}$$

One will notice that in the 4D part of the cost function the future forecasts are now dependent on $\boldsymbol{v}$ as they are obtained by running the model from $\mathbf{x}_{i,(\boldsymbol{v})}^{a}$. Similar to all other some 4D ensemble approaches, this requires only the computation of the

gradient computations can be approximated by the tangent linear model  with the adjoint not being required. In fact all that is required is matrix vector products which can be approximated with finite differences (Tranquilli et al., 2017).

Various 4D-type approximation strategies are also applicable to this cost function extension, though are outside of the scope of this paper.

**4.3 Algorithm**

In practice, instead of dealing with the Gamma distribution parameters of $\alpha$ and $\beta$, we use the parameters $\bar{\upsilon}$ and $\mathrm{Var}(\upsilon)$, such that $\alpha = \frac{\bar{\upsilon}^2}{\mathrm{Var}(\upsilon)}$ and $\beta = \frac{\bar{\upsilon}}{\mathrm{Var}(\upsilon)}$. 
[revised manuscript text omitted]

---

## Author Response (AR2)

**I.   SUMMARY OF CHANGES TO THE TEXT MADE IN RESPONSE TO REVIEWS**

**A.   Summary of changes based on reviewers' suggestions**

1. We have added a citation to our own problems package that was used in the creation of this paper. We have also added rough results about the amount of Lyapunov exponents and fractal dimension of the QGSO model.

2. Additional citations beyond the ones asked for by the referees have been included in order to paint a more complete picture of the current state of knowledge.

3. A sentence has been added thanking the referees for their input into the quality of the paper.

4. The concept of oracles have been clarified a bit and the introduction has been expanded.

5. Some duplicate information has also been removed.

**B.   Summary of changes based on editor's suggestions**

1. Misc: Notation modified once more to use `vec` for vectors.

2. Abstract: added abbreviations.

3. Introduction: Added a citation to a recent optimal experimental design framework for adaptive localization.

4. Section 5: A citation has been added to the model implementations.

5. Section 5.1: The oracle equation has been clarified.

6. Section 5.2.1 and section 5.4: The term 'fractal dimension' has been changed to 'Kaplan-Yorke dimension' as the authors feel that this better reflects the methodology used to obtain the estimates.

7. Section 5.3.1: the interval has been clarified. Language has been changed to be more cautious.

8. Section 5.4: a comma has been removed from the positive Lyapunov exponent estimate.

9. Figures: A title has been added to the multivariate radii choice figure. All figure captions have been truncated to more closely reflect only the information presented, and most of the rest of the caption text has been shifted into the text body. These are too numerous to individually list. Figure 4 has also received a title.

**II.   RESPONSE TO REFEREE 1**

We thank the referee for the many technical suggestions.

**A.   Specific Comments**

*1. The manuscript requires some rearrangement. It is unclear as to why the introduction began by describing a model. This part is more suited for the Background section.*

Authors' response: We agree and have rearranged the text accordingly.

*2. The motivation, objectives, and methods of the study should be clearly stated in the introduction. Some of these are actually discussed in the Conclusions. Please make appropriate revisions.*

Authors' response: The introduction has been extended to better explain the motivation and aims.

*3. Expand your literature review as to include other studies that have done applications of adaptive localization in ensemble methods*

Authors' response: This has been done, in conjunction to the recommendations of referee 2.

*4. P5: the uncertainty in space is represented by $pi(x|y,v)$, but according to equation (12), it should read $pi(x|y,v)$*

Authors' response: We are not sure what this comment is referring to.

*5. P6: for equation (15), you may want to add that the chain rule was also used, in case you want to reach out to students, as derivations might not be too trivial*

Authors' response: We agree, and the modification has been done. An additional comment about symmetric semi-positive definite matrices was also added.

*6. P7: equation (18) takes the form of an EnVar-like cost function. Comment on the applicability of your univariate or multivariate adaptive localization technique for hy- brid EnVar data assimilation systems and also discuss limitations*

Authors' response: An additional comment has been made discussing this.

*7. P9, L3-10: Clarify if univariate localization functions were used and whether the extension of the localiza- tion matrix was done in the present study. Would this help in making your methodology suitable for testing with more complex geophysical models?*

Authors' response: We have clarified that we did not use multiple localization functions in this paper, removing doubt that multivariate localization was used. We are not sure how this point related to more complex geophysical models, however we have clarified that point separately.

*8. P10, L11-17: How do oracles compare to optimal parameter estimation as employed in control theory, with many applications in variational data assimilation?*

Authors' response: A comment was added clarifying that oracles are a type of minimal variance estimator of a full space in a restricted space.

*9. P11, L22: Comment on the choice of 10 members for the ensemble and clarify if varying this number can impact your results*

Authors' response: A comment was added clarifying the choice of 10 ensemble members in Lorenz '96. The question of varying the amount of ensemble members has been explored in other literature, and is outside the scope of this paper.

*10. There are some mathematical notation typos, please revise all equa- tions thoroughly, and make sure to conform to the mathematical notation standards of NPG for all scalars, vectors, matrices, etc.*

Authors' response: All mathematical notation was changed to conform with the NPG guidelines, which mainly meant changing vectors to be boldface italics. The only Exception being the localization matrix $\rho$, as we could not find a way to have non-italic greek letters in conventional latex. Some other notation has been clarified (like indexing into columns using MATLAB syntax) in order to improve readability.

*11. Given the results with a simple geophysical model, provide a brief overview on how this methodology could be tested with a complex numerical weather prediction model (e.g. regional/convection allowing mod- els)?*

Authors' response: A sentence was added in the conclusion about the possibility of applying this approach to the WRF model.

*12. Provide more quantitative estimates in the "Results" and in the "Conclusions" sections*

Authors' response: Quantitative estimates were previously provided in the figures. The figure captions have been changed; the discussion of the results has been included in places where the figures are referenced, with some additional quantitative estimates.

**B.  Technical Corrections**

Authors' response:  All relevant typos have been addressed.  Some comments in the typo section would have erroneously changed the semantics of several key statements and have thus been ignored.

More detailed information was added to the places where the figures are referenced, though we feel full duplication of the information would not be a good presentation of the information in the final publication.

**III.  RESPONSE TO REFEREE 2**

We thank the referee for many detailed suggestions and interesting questions.

**A.  Specific comments**

*Several important references are missing from the Introduction, e.g. Menetrier et al, 2015, Flowerdew, 2015 considered optimal localization, Buehner et al., 2015 suggested scale-dependent localization.*

Authors' response:  These citations have been added.

*I suggest rearranging the text so that everything related to multivariate localization is in section 3.2 (Extension to multivariate localization functions). I think describ- ing univariate case first, and then introducing groups for different localization radii (currently P5, L7-13) when extending to multivariate localization functions might improve the manuscript readability.*

Authors' response:  This is a good point. In order to make the transition smoother and emphasize the fact that the multivariate approach is important, it has been moved into its own section in front of the Bayesian Approach section.

*There seems to be a contradiction between 4.2 and 4.3. P11, L27-28 state that the problem is better suited for multivariate localization, while P12, L11- 12 state that the canonical L96 model is ill suited for multivariate adaptive localization.*

Authors' response:  There is no contradiction between 4.2 and 4.3. To be specific, Lorenz '96 was only tested with univariate adaptive localization; and the prediction was that the problem is not suited for testing univariate adaptive localization. This has been clarified in (the previous) 4.2.3 by adding the word 'univariate'.

*I would like to see more details on the L96 multivariate localization experi- ment setup (Figure 5). Were the groups fixed throughout the experiment? How were they chosen? Did the groups use the same mean and variance parameters at each assimilation cycle? If the groups were fixed, it would be interesting to see how the estimated localization radii vary for different groups throughout the experiment.*

Authors' response:  This relates to the next point about the multivariate Lorenz '96. In the standard Lorenz '96 there is no sensible way to create groupings since all variables have similar dynamics. In the proposed modified Lorenz system we use variable forcing, and essentially create artificial groups of variables driven by similar forcing terms. All this has been clarified in the text.

*I see that in L96 experiments half of the domain is more sparsely observed than the other. Introduction (P2, L34-35) states that optimal localization may depend on observation properties. In L96 experiments, did you see evidence of the optimal localization radii being dependent on observation density?*

Authors' response:  A graph of the multi-group localization radii has been generated and discussed in the text. It illustrates the differences between the optimal localization radii for sparsely observed versus fully observed groups.

*Section 4.4.1, Figure 8. If instead of fixing the mean parameter to be the same as the constant suboptimal radius at each assimilation cycle, the adaptive localization radius estimated at the previous DA cycle was used to estimate the mean parameter, would the adaptive localization radius converge to the optimal one after some DA cycling?*

Authors' response:  The idea of using an online radius mean estimate has been explored; unfortunately this led to suboptimal convergence of the radius estimate, and often to filter divergence. We hypothesize that this is due to the weak correlation between the optimal radii across steps. Exploring this idea further is outside the scope of this paper in our opinion.

**B.    Questions on Extensions**

How does this method extend to the ensemble DA algorithms other than DEnKF?

Authors' response: A paragraph discussing this has been added in the 'Solving the Optimization Problem' section, as the answer is closely tied with that problem.

For large DA applications like NWP, ensemble filters similar to DEnKF typically assimilate observations sequentially, and use $P^f H^T$ localization instead of Schur-product $P^f$ localization which becomes too expensive. Would the method still be applicable in this case, and if so, how would it change?

Authors' response: This is already done in the paper, and we have clarified it in the section discussing this approach.

Do you have a recommendation on how the groups for multivariate adaptive localization should be chosen?

Authors' response: A paragraph discussing this has been added to the multivariate Lorenz '96 localization results.

**C.    Technical comments**

What is subscript $i$ in Equation 10?

Authors' response: This is a typo. The subscript $i$ has been removed from equation (10).

**IV.    RESPONSE TO EDITOR**

We thank the editor for their personal involvement with the article.

Inspecting your figure captions, I find them rather unusual. Except for Fig. 7, you discuss results and sometimes methodology in the captions. Traditionally, figure captions describe what is displayed. They are not meant to present or clarify methodology or evaluate the results. Rev. 1 had a related comment, asking that you discuss results in the text. You responded you did not want duplication. I agree we want to avoid duplication. What I would like to suggest is that you revise your figure captions by moving methodological and evaluation comments to the appropriate places in the text.

Authors' response: All the figure captions have been considerably reduced, and text that is not directly related to the readability of the figure has been moved to the text. Some duplicate information has been allowed to remain as the authors feel that it significantly improves figure readability for 'at-a-glance' figure inspection.

**A.    Responses to additional comments**

1) 2nd sentence of abstract
a) "nearby variables" - should this be replaced by "reference point" or something like that?
b) "should be highly correlated" - did you want to say "should not be..."

Authors' response: The second sentence has been completely rewritten.

2) p3, l.20, not sure if grammar is correct here, consider: "a multivariate variant of which we introduce"

Authors' response: This has been changed.

3) p4, l8 - "an ensemble"

Authors' response: Changed.

4) p5 in original manuscript - I communicated with Rev. 1 who recognized making a typo in her/his comment that s/he clarified: "The uncertainty in state space should be $\pi(x|v, y)$ according to Eq. 12"

*Authors' response:* We believe that no changes need to be made, and that all derivations are correct.

5) p7, l.12 - "in this paper"

*Authors' response:* Changed.

6) p10, l.29 - remove "true"?

*Authors' response:* Changed.

7) p13, l10 - "an ideal"

*Authors' response:* Changed.

8) p15, l.17 - Would this read better? "The largest reduction in error"

*Authors' response:* Changed.

9) p15, l.18 - please revise casual language: "this is leagues more than that of the univariate Lorenz '96"

*Authors' response:* Rewritten.

10) p15, l.27 - a word appears to be missing

*Authors' response:* Will changed to with.

11) p16, l.28 - add comma - "As before, ..."

*Authors' response:* Comma added.

12) p17, l.6 - "reduction in error"; not sure what an "improvement in error" would mean

*Authors' response:* Changed.

13) p18, last sentence of manuscript - poorly constructed with two "and" connectors, can you please revise?

*Authors' response:* The sentence has been revised.

14) First sentence of caption for Fig 6 is awkward, can you please revise

*Authors' response:* All captions have been revised.

[revised manuscript text omitted]

$$m_{\min}(\lambda_i, \lambda_j) \quad = \min\{\lambda_i, \lambda_j\},$$
$$m_{\max}(\lambda_i, \lambda_j) \quad = \max\{\lambda_i, \lambda_j\},$$
$$m_{\mathrm{mean}}(\lambda_i, \lambda_j) \quad = (\lambda_i + \lambda_j)/2,$$
$$m_{\mathrm{sqrt}}(\lambda_i, \lambda_j) \quad = \sqrt{\lambda_i \lambda_j},$$
$$m_{\mathrm{rms}}(\lambda_i, \lambda_j) \quad = \sqrt{\lambda_i^2 + \lambda_j^2}\big/\sqrt{2},$$
$$m_{\mathrm{harm}}(\lambda_i, \lambda_j) \quad = \frac{2\lambda_i \lambda_j}{\lambda_i + \lambda_j}.$$

 Extending this idea to a square-root filter, like the ETKF, would require significant algebraic manipulation, and

account for multiple localization radii associated with different variables as follows:

$$\boldsymbol{\rho} = \left[ m\big(\ell(d(i,j)/r_i), \ell(d(j,i)/r_j)\big) \right]_{1 \le i,j \le n}$$

The localization sub-matrix of state-space variables $x_i$ and $x_j$:

$$\begin{bmatrix} \ell(0) & \boldsymbol{\rho}_{i,j} \\ \boldsymbol{\rho}_{j,i} & \ell(0) \end{bmatrix}$$

is a symmetric matrix for any mean function . When $\ell$ is a univariate compactly supported function, our approach implicitly defines its $m$-based multivariate counterpart.

One of the common criticisms of a distance assumption on the correlation of geophysical systems is that two variables in close proximity to each other might have very weak correlations. For example, in a model that takes into account the temperature and concentration of stationary cars at any given location, the two distinct types of information might not at all be correlated with each other. The physical distance between the two, however, is zero, and thus any single correlation function will take the value one and do not remove any spurious correlations. One can mitigate this problem by considering univariate localization functions for each pair of components $\ell_{i,j} = \ell_{j,i}$, and extending the localization matrix as follows:

$$\boldsymbol{\rho} = \left[ m\big(\ell_{i,j}(d(i,j)/r_i), \ell_{j,i}(d(j,i)/r_j)\big) \right]_{1 \le i,j \le n}.$$

We will not analyze further this extension in the heuristics which are outside the scope of this paper.

**4.2 Adaptive localization via a time-distributed cost function**

We now seek to extend the 3D-Var-like cost function (18) to a time-dependent 4D-Var-like version. As the ensemble is essentially a reduced order model, we do not expect the accuracy of the forecast to hold over the long term as we might in traditional 4D-Var. We thereby propose a limited extension of the cost-function to include a small number $K$ of additional future times. Assuming that we are currently at the $i$-th time moment, the extended cost function reads:

$$
\begin{aligned}
\mathcal{J}_i(\boldsymbol{v}) = \sum_{e=1}^{N} \Bigg[ & \tfrac{1}{2}\left\| \mathbf{S}_{(\boldsymbol{v}),i}^{-1} \mathbf{z}_i^{(:,e)} \right\|_{\mathbf{HP}_{(\boldsymbol{v}),i}^{\mathrm{f}} \mathbf{H}^{\intercal}}^2 \\
& + \tfrac{1}{2}\left\| \mathbf{g}_{(\boldsymbol{v}),i}^{(:,e)} \right\|_{\mathbf{R}_i^{-1}}^2 \\
& + \sum_{k=1}^{K} \tfrac{1}{2}\left\| \boldsymbol{y}_{i+k} - \mathbf{H}\mathbf{x}_{(\boldsymbol{v}),i+k}^{f,(:,e)} \right\|_{\mathbf{R}_{i+k}^{-1}}^2 \Bigg] \\
& + \sum_{j=1}^{g} \left( \beta_i^{(j)} v^{(j)} - \left( \alpha_i^{(j)} - 1 \right) \log\left( v^{(j)} \right) \right).
\end{aligned}
\tag{25}
$$

One will notice that in the 4D part of the cost function the future forecasts are now dependent on $\boldsymbol{v}$ as they are obtained by running the model from $\mathbf{x}_{i,(\boldsymbol{v})}^{\mathrm{a}}$. Similar to all other some 4D ensemble approaches, this requires only the computation of

the gradient computations can be approximated by the tangent linear model with the adjoint not being required. In fact all that is required is matrix vector products which can be approximated with finite differences (Tranquilli et al., 2017).

Various 4D-type approximation strategies are also applicable to this cost function extension, though are outside of the scope of this paper.

**4.3 Algorithm**

In practice, instead of dealing with the Gamma distribution parameters of $\alpha$ and $\beta$, we use the parameters $\bar{v}$ and $\text{Var}(v)$, such that $\alpha = \frac{\bar{v}^2}{\text{Var}(v)}$ and $\beta = \frac{\bar{v}}{\text{Var}(v)}$. For the sake of simplicity we assume that all $v$ are identically distributed, but this is not required for the algorithm to function. The initial guess for our minimization procedure is the vector of means. After minimizing the cost function, the radii for different components will be different. These radii, along with the corresponding localization and $m$ functions, are used to build the localization matrix $\rho$. An outline is presented in Algorithm 1.
* * *
**Algorithm 1** DEnKF Adaptive Localization Algorithm
* * *
**Require:** $[\bar{v}^{(j)}]_j, [\text{Var}(v^{(j)})]_j$

1: $\alpha^{(j)} \leftarrow \frac{\bar{v}^{(j),2}}{\text{Var}(v^{(j)})}$

2: $\beta^{(j)} \leftarrow \frac{\bar{v}^{(j)}}{\text{Var}(v^{(j)})}$

3: $v_0^* \leftarrow [\bar{v}^{(j)}]_j$

4: $v^* \leftarrow \text{argmin}_v \mathcal{J}(v)$ {Eq (22)}

5: $r^* \leftarrow \mathfrak{p}(v^*)$ {Eq (9)}

[revised manuscript text omitted]

---

## Author Response (AR3)

Thank you to the editor and the referees for their patience with this paper.

**I.  RESPONSE TO EDITOR**

I would like to command you for your careful revisions. Based on the second round of reviews and my own inspection, your manuscript is almost ready for publication. I am asking for another "minor revision" as Rev. 1 noted that a bit more substantive interpretation of the results in the Abstract and Conclusions would further help your paper. Both Reviewers also listed a few minor items. I noticed a missing "TO" after "due" on line 4.

Authors' response: The "to" has been added. See response to Referee 1 for the abstract and conclusion.

**II.  RESPONSE TO REVIEWER 1**

*The abstract and the conclusions need to include a more substantial explanation on the findings of the present study.*

The abstract has had some results added to it. A number of sentences were added in the conclusion interpreting the results better.

*P2, L18: do you mean "ensemble of ensemble members"?*

Authors' response: We believe "ensemble of ensembles" is more in line with the terminology that appears in the literature.

*P4, L27: add "the" before "localization matrix"*

Authors' response: "The" has been added.

*P12, L12: is "calculate" the correct word choice in this sentence?*

Authors' response: "calculate" has been changed to "approximately compute" in order to better reflect the actual approach that has to be taken.

*P12, L16: add "the" before "experiments"*

Authors' response: "the" has been added

*P12, L22: change to six-hour*

Authors' response: "six hours" has been changed to "six-hour"

**III.  RESPONSE TO REVIEWER 2**

*page 1, Lines 11 and 17 refer to equations (1) and (2) that are now defined later in the manuscript. Please remove the references.*

Authors' response: The equation references have been removed.

*page 3, Lines 15 and 16. Should $x_i$ be in italics?*

Authors' response: The bold letters have been changed to italics.

*page 9, equations. In (20) and (22), $z^{(:,e)}$ is used, while in (23) $z^{(e)}$ is used. Do those refer to the same thing? If they do, please make the notation consistent through all equations, otherwise please explain in the text what the difference is.*

Authors' response: The gradient equation has been fully fixed. There were a few additional mistakes.

[revised manuscript text omitted]